# Helminth protein enhances wound healing by inhibiting fibrosis and promoting tissue regeneration

Katherine E Lothstein[1], Fei Chen[1], Pankaj Mishra[1], Danielle J Smyth[3], Wenhui Wu[1], Alexander Lemenze[2], Yosuke Kumamoto[2], Rick M Maizels[3], William C Gause[1]

Skin wound healing due to full thickness wounds typically results in fibrosis and scarring, where parenchyma tissue is replaced with connective tissue. A major advance in wound healing research would be to instead promote tissue regeneration. Helminth parasites express excretory/secretory (ES) molecules, which can modulate mammalian host responses. One recently discovered ES protein, TGF-β mimic (TGM), binds the TGF-β receptor, though likely has other activities. Here, we demonstrate that topical administration of TGM under a Tegaderm bandage enhanced wound healing and tissue regeneration in an in vivo wound biopsy model. Increased restoration of normal tissue structure in the wound beds of TGM-treated mice was observed during mid- to late-stage wound healing. Both accelerated re-epithelialization and hair follicle regeneration were observed. Further analysis showed differential expansion of myeloid populations at different wound healing stages, suggesting recruitment and reprogramming of specific macrophage subsets. This study indicates a role for TGM as a potential therapeutic option for enhanced wound healing.

## Introduction

Skin is an important barrier and defense mechanism in mammalian hosts and after an injury, the host must quickly respond with the activation of the wound healing response (1, 2, 3). Wound repair is a highly regulated system that clears the injured area of external pathogens and covers the exposed area to prevent further damage and reduce infection (1, 4). This tissue repair process involves three main overlapping stages for tissue resolution: inflammation, proliferation, and maturation. The inflammatory stage involves the rapid recruitment of multiple cell types to the area of insult immediately after damage. The proliferation phase involves the activation of endothelial cells, macrophages, and fibroblasts that cover the wound. Finally, the maturation phase involves the differentiation and deposition of extracellular matrix (1, 2, 3, 4, 5).

Fibrosis is frequently associated with skin wound repair. It results in tissue scarring in place of regeneration (2) and is considered a significant public health burden, resulting in billions of dollars spent each year (6, 7, 8). The development of a treatment that both rapidly suppresses harmful inflammatory responses and induces the expression of tissue repair factors that favor tissue regeneration over scarring could promote more effective wound healing and lead to better clinical outcomes (2). The helminth-induced type 2 immune response includes properties that enhance tissue repair through control of harmful inflammation, mitigating tissue injury, and also activation of specific molecules that can directly promote wound healing (4, 9, 10, 11, 12, 13, 14). This response may in part be triggered by release of danger signals released by cells damaged during passage of these multicellular parasites through tissues (15). However, helminths themselves can produce a variety of excretory/secretory (ES) products that can modulate host responses and as such potentially have therapeutic significance (16, 17).

These ES molecules so far discovered comprise a number of distinct proteins, carbohydrates, and lipids (16, 17). An ES molecule called TGF-β mimic (TGM), can directly bind the TGF-β receptor in both human and mice (18), ligating the two receptor subunits through the independent N-terminal 3 domains of the protein (19). TGM was originally isolated from an ES supernatant obtained from cultures with the intestinal nematode parasite, *Heligmosomoides polygyrus* yet has no sequence similarity to the TGF-β family (20) and likely has additional functions besides binding the TGFβR. In vitro studies suggest that TGM can induce Foxp3+ regulatory T cells in vitro (21, 22) and TGM exerts anti-inflammatory effects in vivo in models of colitis (23), graft rejection (18), and airway allergy (24). The recent availability of large quantities of recombinant highly purified TGM facilitates in vivo analyses. In this study, we investigated whether administration of the *H. polygyrus* ES product TGM can modulate the progression of skin wound healing.

[1]Center for Immunity and Inflammation, Department of Medicine, New Jersey Medical School, Rutgers, The State University of New Jersey, Newark, NJ, USA   [2]Center for Immunity and Inflammation, Department of Pathology, Immunology, and Laboratory Medicine, New Jersey Medical School, Rutgers, The State University of New Jersey, Newark, NJ, USA   [3]Wellcome Centre for Integrative Parasitology, School of Infection and Immunity, University of Glasgow, Glasgow, UK

Correspondence: gausewc@njms.rutgers.edu

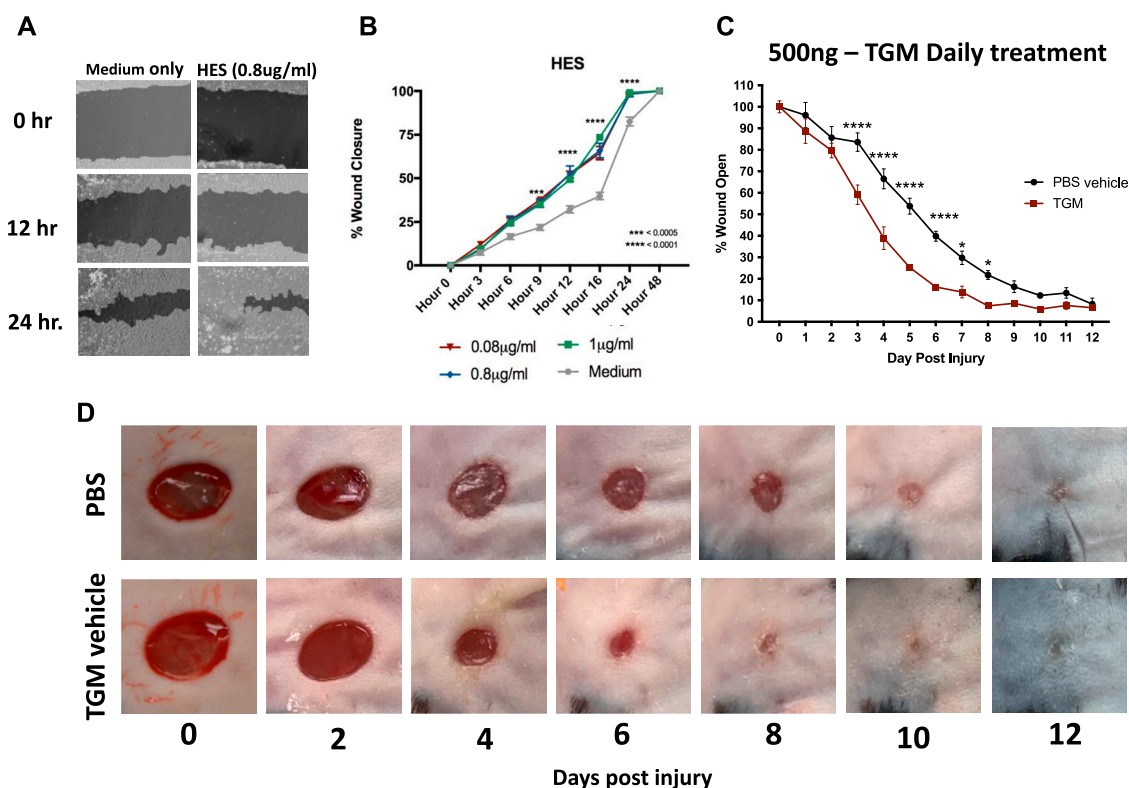

**Figure 1. HES and the purified TGF-β mimic (TGM), accelerates wound closure in both a 2-D scratch test and murine dorsal wounds.**
**(A, B)** A 2-D in vitro scratch test wound model was generated with a 50:50 co-culture of L929 fibroblast and HaCaT keratinocytes to examine the wound closure and migration with the application of HES. **(A)** Representative images, produced from the analysis through the online program, Tscratch, identify the area of open wounds for quantitation (filled-in areas). **(B)** The area of wound remaining open, calculated by Tscratch, was used to quantify the rate of wound closure for different concentrations of HES (0.08–1 μg/ml) compared with media alone from 0 to 24 h. The percentage of wound closure at each time point is compared with the percentage open at hour 0 (**$P$ < 0.01, ***$P$ < 0.001, ****$P$ < 0.0001; n = 5 independent wells per condition). **(C)** 5 mm full-thickness excisional wounds were generated on the dorsal skin of C57BL/6 mice. Wounds were treated with PBS vehicle control or TGM (500 ng) and covered with Tegaderm for the duration of the study. Wound size analysis (ImageJ) was performed on the gross images obtained at each time point during the course of treatment. Treatments were given daily, whereas the dressing was changed every other day. Wound closure rates with either a daily dose of topical 500 ng TGM or PBS vehicle control over 12 d were quantified as the percentage of wound closure at each time point compared with the percentage open at day 0. Results from two or more independent determinations demonstrated similar results (**$P$ < 0.01, ***$P$ < 0.001, ****$P$ < 0.0001; five independent wounds from each treatment were measured through blinded analysis on ImageJ). **(D)** Representative wound images of mice treated topically with PBS vehicle control or TGM (500 ng) on days 0, 2, 4, 6, 8, 10, and 12 demonstrate the rate of wound closure over the course of treatment. **(B, C)** Statistical analysis was performed using a two-way ANOVA test (B, C) with Tukey's multiple comparisons for comparison between all treatment groups at each timepoint. Error bars represent mean ± SEM.

# Results

### Topical TGM treatment accelerated wound closure

Previous studies have suggested helminth infections can trigger immunomodulatory and wound healing effects on surrounding tissues and that associated helminth ES products may contribute to this response (10, 14). To examine specifically whether ES products might promote tissue repair, the effects of ES products from *H. polygyrus* (HES) were analyzed using an in vitro scratch test (25, 26). This test demonstrated an enhanced cell migration by HES compared with media alone (Fig 1A and B). The recent identification of a purified recombinant HES product, TGM, that has an apparent immunomodulatory role (12, 18) and the ability of TGM to bind the TGF-βR (receptor), which has previously been shown to be important in wound healing (27), raised the possibility that TGM may contribute to the effects of HES on wound repair. TGM was thus used in the scratch test used to assess HES activities. As shown in Fig S1A,

administration of recombinant TGM alone showed significantly enhanced wound closure over a 24-h time interval.

Given the accelerated closure in the in vitro wound healing model, we next examined the effects of TGM in a standard murine full-thickness skin wound biopsy model, as previously described (28, 29). A Tegaderm (3 M, St. Paul, MN) bandage was applied over the wound as a fixture to increase skin tension thereby allowing wound closure to occur through re-epithelialization which is more comparable to the human skin wound healing process (28, 29, 30) (Fig S1B and C). TGM was applied as a topical reagent, in a vehicle with 1.5% carboxymethylcellulose in PBS, to generate a more viscous solution that would remain on the wound surface (31). TGM was injected underneath the Tegaderm bandage to directly contact the surface of the wound after a 5 mm biopsy punch was administered to the dorsal skin of the mice (Fig S1B and C). A series of dosing schedules and concentrations demonstrated that there were no significant differences between the various doses when given at day 0 alone, yet identified a daily dose of 500 ng of TGM, as

providing optimal results (Figs 1C and S1D and E). As shown in Fig 1C and D, daily topical application of 500 ng to a 5 mm biopsy wound covered with Tegaderm resulted in significant improvement as early as day 3 with TGM leading to wound closure of 40% compared with 15% when only the PBS vehicle treated control group was administered. The significant effect of daily TGM was maintained between days 4 and 10 with a sustained 30–40% difference in wound closure between the two groups at each time point (Fig 1C and D). By day 10, both treatment groups had closed to a degree in which differences could no longer visually be observed.

Concurrent with the enhanced wound closure, there was a markedly increased wound serous exudate accumulating under the Tegaderm of the TGM-treated wounds, particularly at days 4 and 5 (Fig S2A) (32). LC-MS/MS protein analysis of this liquid demonstrated a significant increase in haptoglobin and thrombospondin-4 in the TGM-treated wounds (Fig S2B and C), which are known to influence macrophage differentiation and epithelial cell migration, respectively (33, 34). Collectively, these studies showed that TGM significantly enhances the rate of wound closure with increases in serous volume and concentrations of factors associated with augmented wound healing.

## TGM increased the granulation of tissue

After wound closure, the maturation phase of wound healing includes the recruitment and activation of various immune cell populations that can influence whether the healing process favors a pro-fibrotic pathway associated with scarring or a more favorable pro-regenerative pathway (35, 36, 37). To examine whether TGM modulates later stages of wound healing resolution, histological analysis of the wound bed was performed after wound closure. Wound tissue granulation is a dermal tissue matrix that replaces the initial clot and fills the wound space with newly formed capillaries, epithelial cells, and infiltrating immune cells (38). TGM-treated wounds showed accelerated granulation formation in comparison to PBS vehicle treated control wounds. Increased granulation tissue was present as early as days 4 and 5 (Fig 2A and B) and was associated with significantly greater wound bed thickness of the tissue by day 5 compared with the PBS vehicle treated control (Fig 2C). This enhanced thickness suggests that TGM is amplifying the formation and increasing the strength of the developing tissue as it heals and matures (38, 39).

Concurrent with the granulation process, the wound edges thicken as keratinocytes are activated to proliferate and migrate over the newly formed dermis that develops from the granulation tissue. This process initiates re-epithelialization (40, 41). With TGM treatment, there was a significant increase in wound leading-edge thickness by day 4, which correlates with the enhanced wound closure that was first observed at this early time point (Fig 2D and E). However, whereas increased granulation and wound closure after TGM administration demonstrate a faster rate of tissue reorganization, these observations do not indicate whether the tissue is following a fibrotic or regenerative developmental trajectory.

## TGM promotes pro-regenerative healing characteristics

Fibrotic development can influence collagen orientation as scar tissue is generally associated with tight parallel collagen bundle alignment, which reduces the skin's elasticity, limiting the tissue's normal function (1, 6, 42). In contrast, uninjured skin tissue collagen exhibits a more cross-linked basket-weave orientation providing greater flexibility to the tissue (43). Quantitative image analysis of picrosirius red-stained tissue sections was used to assess the relative percentage of collagen development in the wound (44). TGM treatment of skin biopsies accelerated collagen deposition by day 7 compared with treatment with PBS vehicle treated control (Fig 3A and B). By day 7, collagen compromised ~50% of the new tissue composition in the TGM-treated wounds, but only 40% of the PBS vehicle treated control skin (Fig 3A and B). However, by day 12, both treatment groups exhibited ~60% collagen composition (Figs 3A and B and S3A). Yet, whereas collagen composition was similar between the two treatments, the collagen orientation was markedly different. On day 12, in the TGM-treated wounds, collagen exhibited the distinct basket-weave morphology characteristic of normal, unwounded skin (Figs 3C and S3A and B) (43, 45). In contrast, PBS vehicle treated control wounds exhibited characteristic scar tissue morphology composed of parallel collagen bundles (Fig 3C).

A fibrotic skin tissue structure cannot support the neogenesis of hair follicles and hair growth. Consequently, hair follicle development is considered an important marker for effective skin tissue regeneration (3, 36, 43). Remarkably, concurrent with the observation of enhanced basket weave collagen morphology, a marked increase in the number of hair follicles were observed in the newly formed dermis of TGM-treated wounds compared with the PBS vehicle treated control wounds treated with the PBS vehicle (Fig 3D and E). At higher magnifications, mature hair follicles were identified along with epidermal projections similar to human rete pegs, which indicate structural strength (43) (Fig 3E).

In addition, in the TGM treated wounds, there was observed complete sebaceous glands and hair shafts, denoting normal skin development associated with tissue regeneration (43) (Fig S3C). In contrast, in the PBS vehicle treated control wounds, the dermal layer remained flat over the epidermis (Fig S3D). Taken together, the increased development of hair follicles and rete peg-like extensions indicates enhanced skin tissue regeneration with reduced scarring following TGM treatment.

$\alpha$-Smooth muscle actin ($\alpha$SMA) expression is another marker used to characterize wound regeneration as it represents myofibroblast activation and differentiation which is critical for wound healing (43, 46). However, excessive myofibroblast activation, which can be quantified through $\alpha$SMA expression, is indicative of scar tissue development (43, 46). $\alpha$SMA expression within the wound was quantified through the analysis of stained frozen skin tissue samples collected on days 7 and 12 after skin biopsy. Analysis of the fluorescent intensity of the immunofluorescent stained slides, demonstrated that $\alpha$SMA expression increased after day 0 and was similar in both TGM treated and PBS vehicle treated control groups on day 7 (Figs 3F and G and S3E). However, by day 12, elevations in $\alpha$SMA were significantly less marked in the TGM-treated wounds compared with the PBS vehicle treated control (Fig 3F and G). Furthermore, gene expression analysis of skin wounds by qRT-PCR of *Acta* ($\alpha$SMA) levels

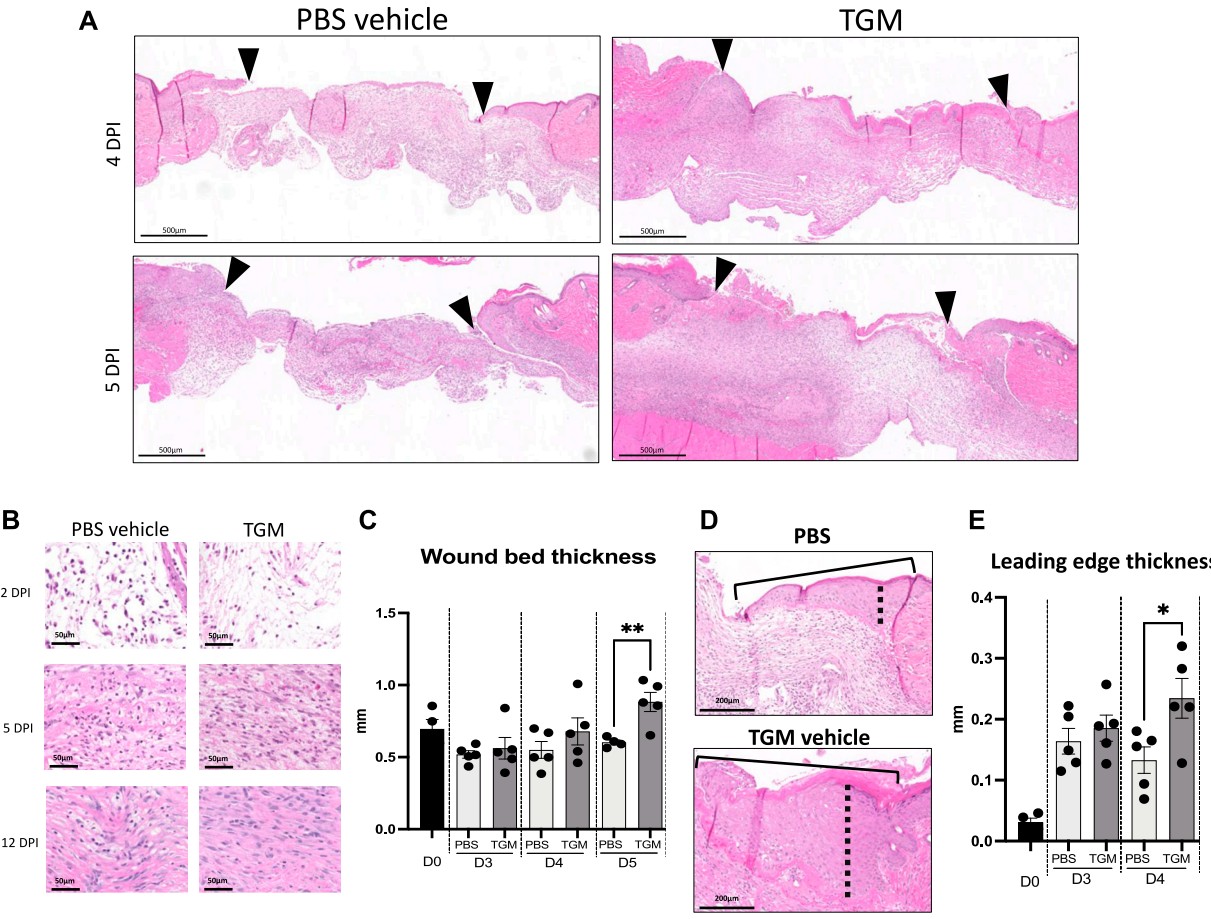

**Figure 2. TGF-β mimic (TGM) enhances the granulation development and thickness of the epidermis/dermis within the newly formed wound.**
**(A)** Representative hematoxylin and eosin (H&E) stained images on days 4 and 5 post-injury. Eosin staining was used to identify granulation tissue formation within the wound beds (marked by arrows) of the TGM and PBS vehicle control wounds at each timepoint. **(B)** Representative H&E images at higher magnification (20x) were used to visually assess the cellular and extracellular matrix deposition within the wound beds of PBS vehicle control and TGM-treated wounds on days 2, 5, and 12 post-injury. **(C)** H&E stained wounds were used to measure the wound thickness of the treatment groups on days 3, 4, and 5 post-injury. **(A)** Blinded analysis in ImageJ was used to measure the thickness of the granulation tissue within the wound bed (area between the black arrows in (A)). Multiple lengths obtained from one wound bed were averaged. Results from two or more independent determinations demonstrated similar results (**$P < 0.01$; five biologically independent samples were used per treatment for the blinded analysis on ImageJ). **(D)** Representative H&E images of the wound edge, marked by the solid black line, were used to measure wound edge thickness. Blinded analysis in ImageJ was used to measure the thickness of the wound edge. The dotted black line indicates the length measured. **(E)** H&E stained images were used to measure the thickness of the leading edge within the wound bed. Measurements for PBS vehicle control and TGM were analyzed on days 0, 3, and 4. The thickest part of the wound edge over the granulation tissue was used for the analysis. Results of two or more independent determinations demonstrated similar results (**$P < 0.05$; five biologically independent samples were used per treatment for the blinded analysis on ImageJ). **(C, E)** Statistical analysis was performed using a $t$ test (C, E) to compare the two treatments at each time point. Error bars represent mean ± SEM.

confirmed attenuated expression by myofibroblasts in TGM-treated wounds on day 12 (Fig 3H) (46). Together, these data suggest suppression of myofibroblast activation which may contribute to reduced scarring in wound beds after TGM treatment. Collectively, these histological observations indicate that daily TGM application, in addition to accelerating wound closure, enhances pro-regenerative relative to pro-fibrotic tissue wound healing.

## TGM-induced wound closure requires TGF-β receptor binding domains

TGF-β plays a role in multiple mechanisms associated with wound healing, which are initiated through binding to the heterodimer of TGF-βRI and II (1, 47, 48). Since TGM also binds to this receptor complex, we tested the possible role of wound healing modulation through the specific TGF-βR binding activity of TGM. TGM is composed of 5 domains, with domains 1-3 (TGM D1-D3) containing the TGF-βR binding activity, and domains 4 and 5 (TGM D4-D5) likely involved with additional, yet currently unknown, binding activities (Fig 4A) (20). Wound biopsies were treated daily with 500 ng of truncated recombinant TGM constructs of TGM D1-D3, or TGM D4-D5, as well as TGM D1-D5 (full-length TGM), and PBS vehicle treated control for 7 d (5 mice/treatment group). The gross image area measurement, compared with day 0, at each time point, demonstrated that TGM D1-D5 (500 ng, n = 5), as seen in our earlier experiments, led to significant wound closure compared with the PBS vehicle treated control (n = 5) on day 4 with 50% of the wound closed compared with the 30% closed in the PBS vehicle treated

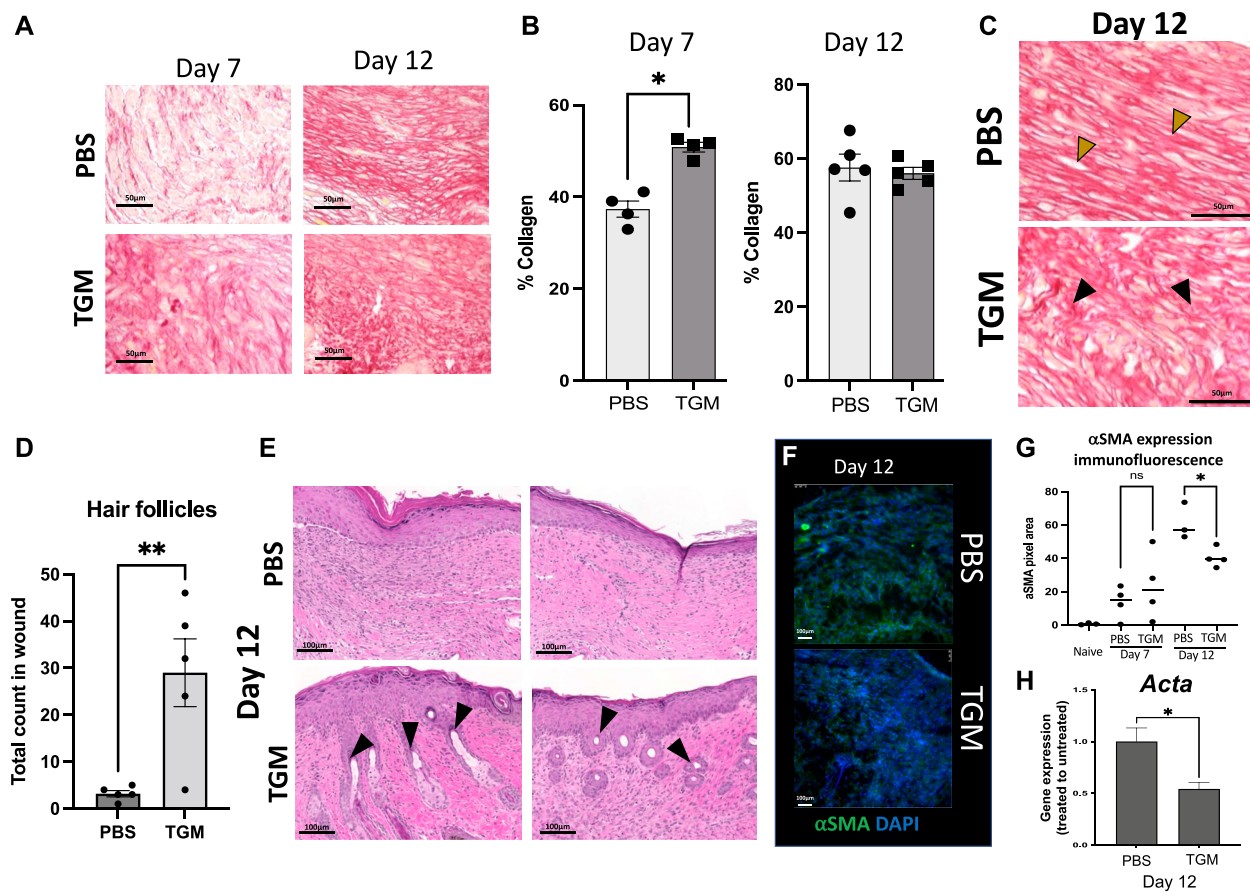

**Figure 3. TGF-β mimic-treated wounds are associated with pro-regenerative collagen deposition and orientation, enhanced hair follicle frequency and regulated myofibroblast expression.**

**(A)** Picrosirius red stain was used to identify the orientation and quantify the percentage of collagen within the newly formed wound bed. Representative images of picrosirius red stain for collagen quantification (20x) identify the amount of collagen in the two treatments on days 7 and 12. **(B)** The collagen area within the 20x picrosirius red-stained slides was quantified using the MRI Fibrosis Tool plugin on ImageJ. Multiple images were taken for each sample spanning the length of the wound bed and then averaged for that sample. Results from two or more independent determinations demonstrated similar results (**P < 0.01; five biologically independent samples were used per treatment for the blinded analysis on ImageJ). Statistical analysis was performed using a t test to compare the two treatments at each time point. Error bars represent mean ± SEM. **(C)** Representative images of picrosirius red-stained slides on day 12 that were used to identify the collagen orientation in PBS vehicle control and TGF-β mimic treated wounds. Yellow arrows highlight examples of collagen in a parallel orientation. Black arrows highlight examples of basket weave collagen deposition. Images represent similar images among the other five samples in each treatment. **(D)** H&E stains were used to quantify the frequency of hair follicles within the wound beds on day 12. Hair follicles were included in the count if they were located within the wound bed and beneath the thickened epidermis which represented the area of the wound. (**P < 0.01; five biologically independent samples were used per treatment for the blinded analysis on ImageJ). A t test was performed to compare the two treatments at each time point. Error bars represent mean ± SEM. Results from two or more independent determinations demonstrated similar results. **(E)** Representative images of H&E stained slides from wounds on day 12 were used to identify skin maturation through the appearance of hair follicle formation within the wound bed. Black arrows = hair follicles. **(F, G)** Alpha smooth muscle actin (αSMA) representing myofibroblast formation, was quantified in all images on day 12 through immunostained slides. **(F)** Representative immunofluorescent stained images of wound beds stained with αSMA on day 12; αSMA (green fluorescent signal), DAPI (blue fluorescent signal). **(G)** Immunostained slides with αSMA were quantified on days 7 and 12 by the fluorescent intensity measured as pixel area using ImageJ. (*P < 0.05; five biologically independent samples were used per treatment for the blinded analysis on ImageJ). Results from two or more independent determinations demonstrated similar results. **(H)** Gene expression analysis of Acta2 expression was used to further analyze the αSMA expression within the wound bed on day 12 post-wound. (*P < 0.05; five biologically independent samples were used per treatment for the blinded analysis on ImageJ). Results from two or more independent determinations demonstrated similar results. **(F, G)** Statistical analysis was performed using a t test (F, G) to compare the two treatments at each time point. Error bars represent mean ± SEM.

control. TGM D1-D3 (500 ng, n = 5) showed almost identical wound closure activity compared with TGM D1-D5 (Fig 4B and C). However, TGM D4-D5 (500 ng, n = 5) did not show enhanced wound healing relative to the PBS vehicle treated control treatment (Fig 4B and C). Both TGM D1-D5 and TGM D1-D3 showed significant wound closure compared with TGM D4-D5 as early as days 3 and 4. These findings demonstrate that the wound healing activity of TGM is mediated specifically through TGF-βR binding.

## TGM treatment reprograms macrophages

Different subsets of macrophages are associated with different stages of wound healing (2, 9). To assess whether TGM modulated macrophage activation and recruitment, flow cytometric analysis of myeloid cell populations in the wound bed was performed. In the PBS vehicle treated control group, the macrophage population (CD11b+ F4/80+ CD64+), as a percentage of all CD45+ cells, was

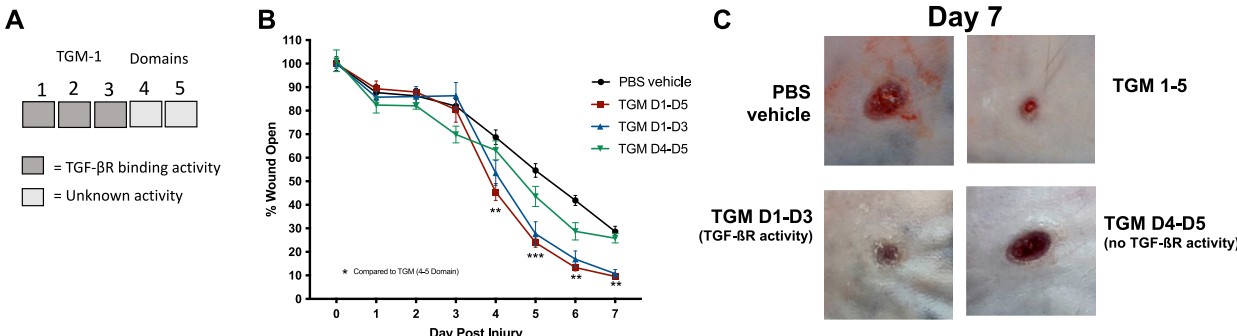

**Figure 4. In vivo wound biopsy with truncated variants suggest TGF-β mimic (TGM) enhances wound healing through TGF-ßR activity.**
**(A)** The TGM molecule contains 5 domains. Domains 1 through 3 comprise the TGF-ßR domain activity. The activity of domains 4 and 5 is currently unknown. **(B)** 5 mm full-thickness excisional wounds were generated on the dorsal skin of C57/Bl6 mice. Wounds were treated with PBS vehicle control or whole TGM (TGM D1-D5; 500 ng) or TGM containing only domains 1 through 3 (TGM D1-D3; 500 ng) or domains 4 and 5 (TGM D4-D5; 500 ng) and covered with Tegaderm for the duration of the study. Wound size analysis was performed on the gross images obtained at each time point during the course of treatment. Treatments were given daily whereas the dressing was changed every other day. Wound closure rate with either a daily dose of topical TGM D1-D5, TGM D1-D3, TGM D4-D5, or PBS vehicle control over 7 d was quantified as the percentage of wound closure at each time point compared with the percentage open at day 0. (**$P < 0.01$, ***$P < 0.001$, ****$P < 0.0001$; black stars represent the significance of TGM D1-D5 compared with TGM D4-D5; five independent wounds from each treatment were measured through blinded analysis on ImageJ). Statistical analysis was performed using a two-way ANOVA with Tukey's multiple comparisons for comparison between all treatment groups at each timepoint. Error bars represent mean ± SEM. **(C)** Representative wound images of mice treated topically with PBS vehicle control, TGM D1-D5, TGM D1-D3, or TGM D4-D5 (500 ng) on day 7 provide a visual comparison of the area of wound remaining open between the four different groups. Results from two or more independent determinations demonstrate similar results.

decreased at 1–2 d post-injury; however, the percentage began to increase and peaked between 5–7 d (Fig 5A and C, shown in beige), similar to previously published results (49). However, after TGM treatment, macrophages peaked earlier at day 3, consistent with increased wound closure at early timepoints (Fig 6B and C). From days 2 to 5, macrophages comprised a greater percentage of CD45⁺ cells in TGM-treated mice relative to the mice treated with the PBS vehicle treated control (Fig 5B and C). By day 7, the TGM-treated macrophage population decreased whereas their counterparts in PBS vehicle treated control tissues increased significantly (Fig 5C).

During the wound healing transition from the proliferative phase to the maturation phase, alternatively activated (M2) macrophages, expressing CD206, expand and are thought to play a crucial role in the maturation stage of the wound healing process (49, 50, 51), typically located at the leading edge of the wound where they interact with keratinocytes (52). In naïve skin, CD206+ macrophages are abundant, representing ~80% of all tissue-resident macrophages, as demonstrated in our unwounded samples (Fig 5D) (49). To test the effect of TGM on the frequency of this specific macrophage subset, CD206+ macrophages were assessed on days 2, 4, and 6 after punch biopsy. PBS vehicle treated control mice showed a significant decrease in the percentage of CD206+ macrophages (CD11b+ F4/80+ CD64⁺) by day 2 that was largely restored by day 3 (Fig 5D), consistent with previous studies (49, 53). In contrast, in TGM treated wounds, the CD206+ population decreased in the TGM treated group by day 2, but as wound healing progressed, increases in the CD206+ macrophages were delayed until day 7 (Fig 5D). In both groups, as a percentage of total CD45⁺ cells, there was a peak accumulation of CD206+ macrophages on day 7, which was still significantly reduced in the TGM-treated wounds (Fig 5E).

### TGM induces a CD206-wound repair macrophage population

Our studies indicated marked changes in macrophage phenotypes because of TGM application after skin wounding. The wound healing response generates a type 2 immune pathway that activates CD206+ M2 macrophages in the presence of IL-4 and IL-13 (54, 55). To test whether TGM directly affects macrophage activation, bone-marrow derived macrophages were stimulated, for 16 h, with M-CSF and IFNγ/LPS or IL-4/IL-13 (Fig 5F and G). As expected, only a small percentage (20%) of the cells treated with IFNγ/LPS expressed CD206, and the addition of TGM did not significantly affect the expression of this marker (Fig 5F). In contrast, culture with IL-4 and IL-13 resulted in a marked increase in CD206+ macrophages (Fig 5G). However, unexpectedly, the application of TGM significantly reduced the expression of CD206+ macrophages (Fig 5G). These results indicate that TGM can directly influence macrophage activation in the context of the specific cytokine environment.

The reduced frequency of CD206+ macrophages concurrent with the enhanced wound healing in TGM-treated wounds appeared contradictory with previous studies suggesting a role for this macrophage subset in tissue repair (49). To further analyze the potential heterogeneity in the activation of immune cells in the wound bed following TGM treatment, we performed single-cell RNA sequencing (scRNAseq) on CD45⁺ purified cells isolated from three biological replicates 3 d after wounding. Analysis of the six samples revealed 10 distinct clusters that were composed primarily of macrophage or monocyte populations (56, 57, 58) (Figs 6A and B and S4A and B). Clusters 0, 1, 2, and 5 resembled macrophages with their increased expression of *Adgre1* (F4/80) and *Itgam* (CD11b) (Figs 6C and S4C) (56). Several other significant identified populations included neutrophils (Cluster 3) (56), natural killer cells, Langerhans cells (Clusters 7), and dendritic cells (Clusters 4, 8, and 9) (Figs 6A and B and S4C) (56).

To further dissect potentially more distinct macrophage subsets within these broadly defined clusters, we analyzed the differential gene expression of cells within the monocyte/macrophage clusters 0, 1, 2, and 5 between PBS vehicle treated control and TGM treated wounds. A considerable difference was detected in gene expression

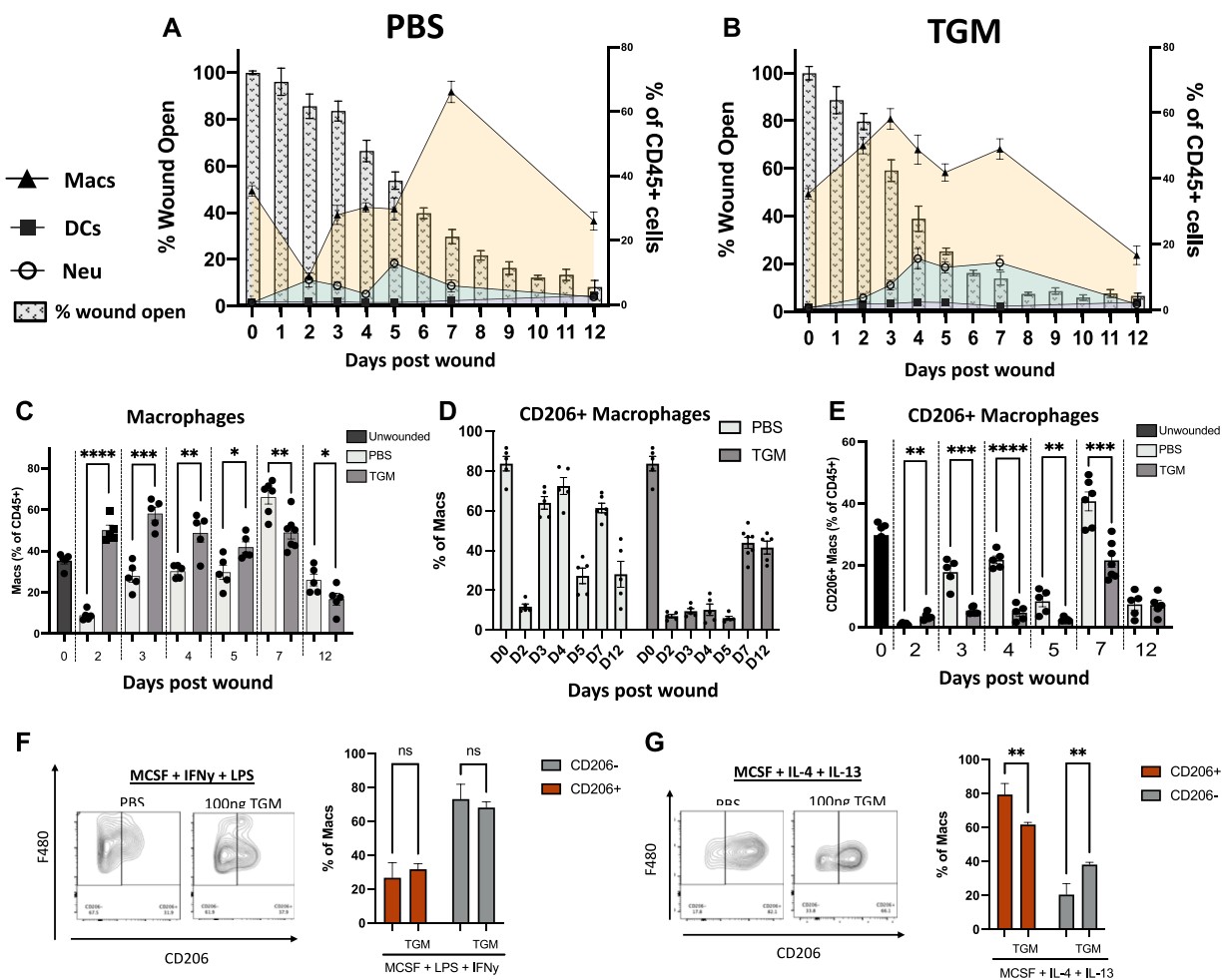

Figure 5. **TGF-β mimic (TGM) treatment reprograms myeloid cell expansion and delays macrophage CD206 expression.**
**(A, B)** Flow cytometric analysis of the cell population within the wound beds from day 0 to day 12 for PBS vehicle control (A) and TGM (B) treated wounds. The analyzed cells include macrophages (CD11b+ F480+ CD64$^+$), dendritic cells (MHCII+ CD11c+ CD64$^-$) and neutrophils (CD11b+ Ly6G+ CD64$^-$) as a percentage of the total CD45$^+$ cells. The cell populations at each time point are overlaid on a bar graph representing the wound closure of each treatment group. Left Y-axis denotes the percentage of myeloid cells of total CD45$^+$ cells represented in the line graph. Right Y access denotes the percentage of wound open as represented in the background bar graph. The X-axis represents the days post wounding. **(C)** Flow cytometric analysis of the frequency of macrophages (CD11b+ F480+ CD64$^+$), as a percentage of all CD45$^+$ cells, in PBS vehicle control and TGM treated wounds from day 0 to day 12. (*$P < 0.05$, **$P < 0.01$, ***$P < 0.001$, ****$P < 0.0001$; five biologically independent samples were used per treatment at each time). Representative results from two or more independent experiments are shown. **(D)** Flow cytometric analysis for the frequency of the CD206+ macrophages as a percentage of total macrophages between PBS vehicle control and TGM treated wounds. Representative results from two or more independent experiments are shown. **(E)** Flow cytometric analysis of the frequency of CD206+ macrophages as a percentage of total CD45$^+$ cells in PBS vehicle control and TGM treated wounds from day 0 to day 12. (*$P < 0.05$, **$P < 0.01$, ***$P < 0.001$, ****$P < 0.0001$; five biologically independent samples were used per treatment at each time point). Representative results from two or more independent experiments are shown. **(F, G)** Flow cytometric analysis of the frequency of CD206+ macrophages gated on CD11b+ F4/80+. **(F, G)** BMDMs were isolated and stimulated for 16 h in a (F) classically activated (LPS/IFNy) macrophage-inducing environment or (G) alternatively activated (IL-4/ IL-13) macrophage environment with or without TGM. Bar graphs represent the frequency of CD206+ or CD206- macrophages as a percentage of all macrophages (*$P < 0.05$, **$P < 0.01$, ***$P < 0.001$, ****$P < 0.0001$; three wells were used per treatment). Statistical analysis of the CD206+ or CD206– populations was performed using a one-way ANOVA test with Tukey's multiple comparisons for comparison between all treatment groups. Error bars represent mean ± SEM.

profiles between TGM- and PBS vehicle treated control cells within the defined macrophage clusters (Figs 6D–F and S4D). Specifically, there was reduced expression of *Mrc1*, which encodes CD206 (56), in the TGM-treated compared with PBS vehicle treated control groups, although significant increases were observed in other M2 markers, including *Arg1* and *Fn1* (Figs 6D–F and S4D) (56, 59). Furthermore, another M2-associated marker, *Chil3*, was increased in TGM-treated samples in clusters 0, 2, and 5 (Figs 6D–F and S4D). Other markers associated with the M2 macrophage phenotype, including *C1qa, Cc18,* and *Folr2* (56) and M2 markers associated with

skin regeneration, fibronectin (*Fn1*) and *Ecm1* (59) (Figs 6D–F and S4E), were also found to be increased in wound biopsies treated with TGM further confirming reprogramming of the macrophage population in response to TGM. Notably, *Mrc1*/CD206 expression was abrogated in all macrophage clusters, with concomitant up-regulated expression of other M2-associated genes, suggesting a selective loss of this marker rather than the outgrowth of a specialized CD206-negative subset. These findings suggest macrophages expressing wound healing associated markers such as *Arg1* and *Chil3* may not require CD206 expression to be identified as a M2

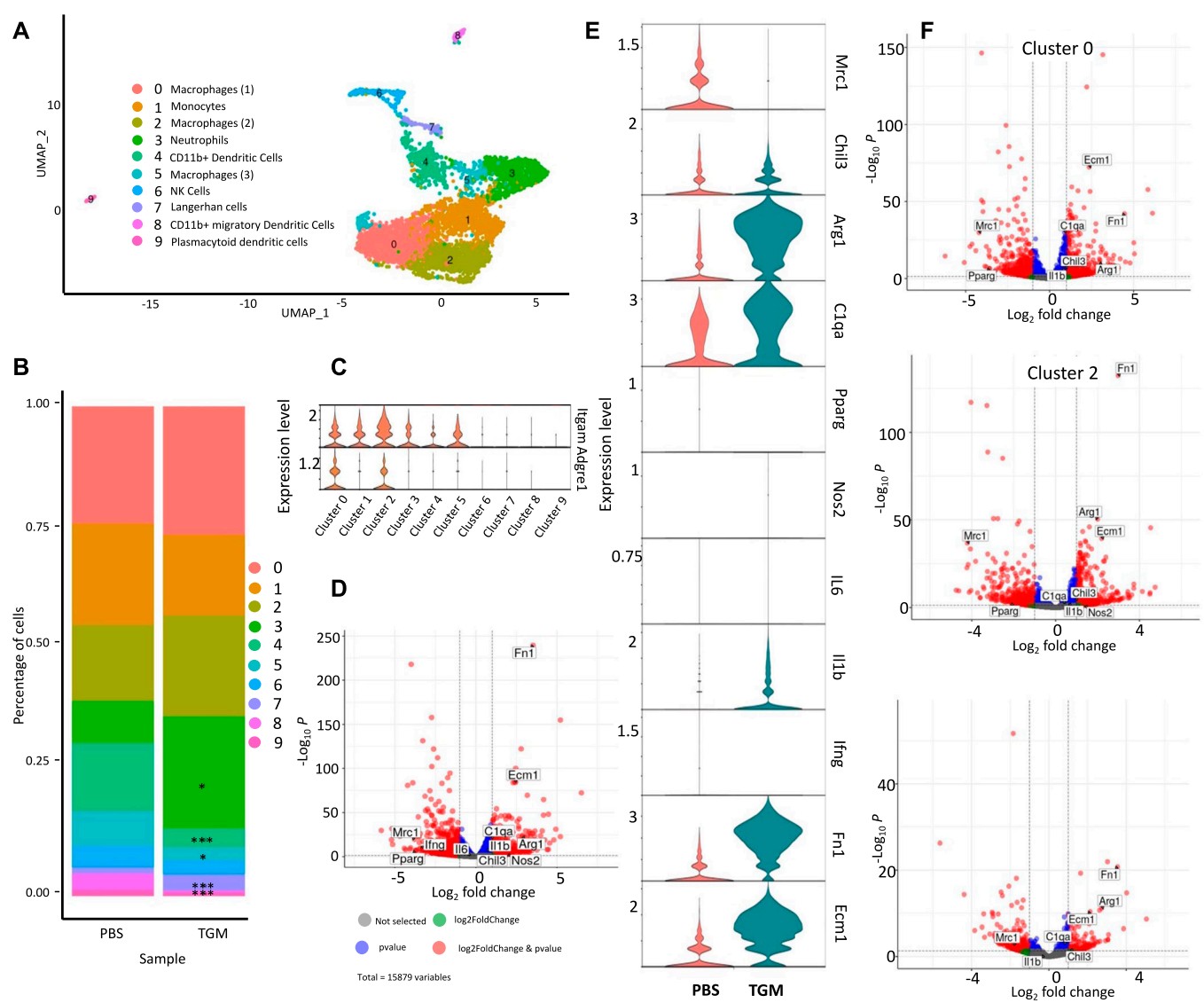

**Figure 6. TGF-β mimic (TGM)-treated macrophages have reduced CD206+ expression yet are enriched for other tissue repair-associated markers.**
scRNAseq analysis of CD45⁺ leukocytes obtained from wound bed at day 3 after skin injury. Skin wounding was performed as described in Fig 1. **(A)** A uniform manifold approximation projection plot of the single cells obtained from CD45⁺ cells purified from PBS vehicle control and TGM treated wounds on day 3 after skin injury. 10X genomics scRNAseq using was performed on the CD45⁺ purified cells treated with PBS vehicle control or TGM (three biologically independent samples were used per treatment labeled with three unique TotalSeq Hashtag Antibody markers). The analysis identified nine distinct cell clusters. **(B)** The percentage of single-cell distribution among the clusters between the PBS vehicle control and TGM treated samples. Asterisks represent the significant difference in the cluster population size between the two treatment groups (*P < 0.05, ***P < 0.001). **(C)** Violin plots represent the differential gene expression of macrophage markers (*Itgam* and *Adgre1*) within the clusters of TGM to PBS vehicle control combined. **(D)** Volcano plot representing the differential gene expression of the monocyte/macrophage populations between PBS vehicle control and TGM treated wounds (clusters 0, 1, 2, 5 combined). Horizontal dotted lines represent significant fold changes of expression for each treatment group. Vertical dotted lines represent a significant *P*-value (<0.05). Red dots represent genes significantly up-regulated within the two treatment groups. **(E)** Violin plot further define the macrophage populations by showing the differential gene expression of alternatively activated (M2) macrophage markers (*Arg1, Chil3, Mrc1*) and classically activated (M1) macrophage markers (*Nos2, IL6, IL1b,* and *Ifng*) within 0, 1, 2, 5 combined. **(F)** Volcano plots represent the differential gene expression of TGM to PBS vehicle control (clusters 0, 2, and 5). Horizontal dotted lines represent significant fold changes of expression for each treatment group. Vertical dotted lines represent a significant *P*-value (<0.05). Red dots represent genes significantly up-regulated with the two treatment groups.

macrophages. It will be important in future studies to examine whether these changes observed in expression of M2-associated genes also occur at the protein level.

By contrast, markers typically associated with classical inflammatory macrophages (*IFNγ, IL-6, Nos2*) were minimally expressed in all clusters 1 through 5, with the exception of *IL-1β* which was amplified particularly in cluster 2 (Figs 6E and S4D). These data show that TGM alters the macrophage landscape in tissues undergoing wound healing responses. Taken together, these analyses suggest that macrophages induced following TGM treatment generally show a reduced expression of CD206, but up-regulation of many markers associated with M2 macrophages and the wound healing process.

# Discussion

Cutaneous wound healing is a complex process that mitigates tissue damage and restores the integrity of damaged barrier surfaces. Rapid wound closure following injury is essential in reestablishing a protective barrier against infection and further injury (4, 40). However, it can also favor pro-fibrotic development over pro-regenerative mechanisms. Fibrosis and associated scarring can generate many long-term harmful effects that may potentially be avoided with improved treatments. Recent studies suggest that the later stages of tissue repair may be sufficiently plastic that a pro-fibrotic or pro-regenerative process may be favored depending upon the cytokine and cellular environment (35, 36, 37). In this study, we have developed a novel therapy for the treatment of cutaneous wounds that favors regenerative wound healing over fibrosis and scarring. We administered the recently identified recombinant TGM molecule, originally isolated from helminth ES products, immediately underneath a protective Tegaderm bandage. Our findings showed enhancement of wound closure and marked increases in tissue regeneration as noted by re-epithelialization, collagen crosslinking, and hair follicle regeneration. This finding presents the possibility of TGM playing a role in activating the stem cells found within the hair follicles, as Tregs have been shown to enhance the proliferation of these cells (36, 60). Our studies further showed modulation of immune cell composition and activation at the tissue repair interface, including reduced expression of CD206 by macrophage populations expressing markers important in wound healing and characteristic of M2 macrophage activation. TGF-$\beta$ is known to play an important role in wound healing and has previously been shown to promote M2 macrophage activation (43, 47). Our findings now indicate that TGM also promotes activation and proliferation of tissue-repair associated M2 macrophages through either direct or indirect mechanisms.

Extensive research in wound healing treatments has produced a number of useful but as yet suboptimal therapies. Studies with chronic wounds or burns have used negative pressure or oxygen therapy to enhance the wound healing process whereas other techniques involve the use of skin transplantation. However, whereas successful in enhancing wound closure, many times they fail to replicate the normal skin morphology and require constant maintenance to protect the new skin (61, 62). Early studies on the effect of the application of exogenous TGF-$\beta$ directly to the wound were inconclusive. Some studies suggest that TGF-$\beta$ applied topically had little to no effect whereas others observed possible inhibition of endogenous TGF-$\beta$ activity (63). Therefore, the focus of research has shifted to other treatment options. More recent technology using topical stem cell therapy has demonstrated effectiveness; however, the maintenance of these cells and the ability to standardize the methods for patients have been a major obstacle (61, 64). For standard wound care following postoperative wounds or wounds without underlying diseases, dressings have been the preferred practice of care to both prevent infection and enhance the wound closure (62). The application of hydrogels or various growth factors has been used in combination with these bio-occlusive covers. Yet, whereas showing promising outcomes, the long-term effects of the sustained application of these growth

factors have led to the development of severe side effects (65). Furthermore, many of the studies performed with current prospective wound treatments have been primarily in in vitro settings or focused on extensive burns or wounds with underlying diseases (61). Our findings now show that topical application of TGM is effective in both enhancing wound healing and generating an overall beneficial long-term outcome for standard wound treatment. In addition, as a recombinant agent, TGM is easily produced and can be standardized for treatment in combination with a bio-occlusive bandage, such as Tegaderm.

TGM has many other characteristics that may make it a viable option for therapeutic use. Although sharing the capacity to specifically bind the TGF-$\beta$R complex, TGM is structurally distinct from TGF-$\beta$ and, in fact, shares more similarities with the Complement Control Protein family (19, 20). Furthermore, possibly because of its large size, it can bind TGF-$\beta$RI and TGF-$\beta$RII sites that are well-separated and not directly adjacent to one another as in the TGF-$\beta$ receptor complex. TGM is also more bioactive than TGF-$\beta$ (20). In addition, and perhaps more significantly for therapeutic use, TGM does not require proteolytic cleavage for activity, shows reduced immunogenicity, and it is more stable and easier to modify for pharmacological use (18, 66). Taken together, these studies provide a significant framework for the potential use of a highly purified helminth ES product as a therapy to promote cutaneous wound healing.

# Materials and Methods

### Cell culture

A 2-D cell wound healing model system was performed using a dermal fibroblast cell line, L929, and a keratinocyte cell line, HaCaT. These cells were cultured in DMEM (Thermo Fisher Scientific) supplemented with 4 mM L-glutamine (Gibco), 1 mM sodium pyruvate (Gibco), 10% FBS (Gibco), and antibiotics (1% anti-anti; Gibco) at 37°C in an atmosphere containing 5% $CO_2$ as previously described (26). Cells were expanded at 70–80% confluency.

### Scratch test analysis

Scratch test analysis with SMA, TSES, HES, and TGM were performed as previously described (12, 18, 26). Briefly, 24 well tissue culture plates were pretreated with 0.2 mg/ml of collagen from rat tail (1 mg/ml stock diluted in 0.1 M acetic acid; Sigma-Aldrich) for 2 h at 37°C and 5% $CO_2$. After incubation, the plates were rinsed with warm sterile PBS. Each well was seeded with 100,000 cells in 1.2 ml of culture media. The wells contained keratinocytes alone, fibroblasts alone, or both cell types at a 50:50 ratio. The cells were incubated at 37°C and 5% $CO_2$ for 24 h to generate a confluent monolayer (~70–80% plate coverage). Upon confluency, each well was scored with a p1000 pipette. The media was immediately removed to collect the dislodged cells and replaced with fresh cell culture media. A marker was used on the plate underside to mark the open area to aid as an hour 0 control for the analysis over time. Two photos from each well were taken on an EVOS Core Imaging System

at time 0 right after the scratch and additional photos were collected at 3, 6, 9, 12, 18, and 24 h. The online program, Tscratch, was used to measure the area of the open wound (67). The percentage of wound closure was calculated by the area of the wound area at a given time compared with the wound area at hour 0. Each treatment group was performed in quintuplicate.

## Preparation of TGM

Purified recombinant *Hp*-TGM was generously provided by Dr. Rick Maizels' lab at the University of Glasgow and prepared as previously described (18, 20). Briefly, the TGM was expressed in mammalian tissue culture cells; as such there was no bacterial involvement and no contamination with LPS or any other bacterial products. The TGM is expressed at a high level and elutes as a single peak from a metal chelating column via the His tag. The lyophilized TGM recombinant protein was resuspended in sterile $H_2O$ and 35% of sterile glycerol. The TGM was aliquoted into 10 $\mu$l or greater volumes for storage in the −80°C.

## Skin biopsy wound healing model

C57/BL6 mice were purchased from Jackson Laboratory (stock # 000664). For most of the wound healing studies, 8–9 wk old male mice were used as there are morphological differences between male and female mice that might impact the wound healing response (68). The mice were maintained in a pathogen-free facility at the Rutgers New Jersey Medical School Comparative Medicine Resources. The wound healing studies used a revised protocol based upon a standard procedure (30, 33). Briefly, 2 d before the wounding, the mice were anesthetized with a rodent cocktail containing Ketamine (80 mg/kg) and Xylazine (10 mg/kg) and their dorsal skin was shaved followed by a depilatory cream (Nair) to further remove hair. On the day of the surgery, the mice were again anesthetized with the ketamine/xylazine cocktail. Following a three-step betadine/ethyl alcohol wash, two full-thickness wounds (epidermis, dermis, and subcutaneous) were induced on each side of the midline dorsal skin of the mice using a sterile 5 mm biopsy punch (Integra Miltex). A sterile transparent bio-occlusive film (3 M; Tegaderm) was applied over the wound (29, 33). The mice were split into two treatment groups with one group receiving a PBS vehicle treated control (50 $\mu$l) and the other receiving TGM (500 ng). A stock of 1.5% carboxymethylcellulose in sterile PBS was made. TGM was diluted in this vehicle to generate a 500 ng of TGM/50 $\mu$l of solution. The 1.5% carboxymethylcellulose/PBS solution (±TGM) was injected through the Tegaderm and on top of each wound (50 $\mu$l/wound). The mice were allowed to recover from the procedure in a warming chamber and then house separately for the duration of the study. Mice were anesthetized daily with isoflurane to image the wounds and apply an additional 50 $\mu$l of the 1.5% carboxymethylcellulose/PBS vehicle treated control to each wound with or without 500 ng of TGM at the appropriate time point. The Tegaderm was replaced every other day. The wounds were digitally photographed daily until the completion of the study. The images were blinded through the Blind Analysis plugin on Image J (NIH). They were then measured and analyzed through Image J. A ruler placed beneath the wound for each image was used to normalize and measure the area of each wound in the Image J program. The percentage of wound closure was calculated by the area of the wound at a given time compared with the wound area at Day 0. Multiple images were collected for each mouse wound and then averaged. On the day of harvest, the two wounds and surrounding tissue from each mouse were collected with 10 mm biopsy punches (Acuderm Inc; Thermo Fisher Scientific). One wound biopsy was used for further flow cytometric and gene expression analysis. The other wound from the same mouse was halved with a scalpel for use in immunofluorescence and histological analysis (explained below). Studies using the truncated TGM variants were performed following the same protocol as the whole TGM treatment studies. All in vivo studies were performed in accordance with a protocol approved by the Institutional Animal Care and Use Committee (IACUC) at Rutgers University.

## Flow cytometry

Single cell isolation and flow cytometric analysis of the skin was performed as previously described (69). Excised skin was placed in a 0.5% Dispase II solution (diluted in PBS; Roche) and placed at 4°C overnight. The following day, the digested skin was minced and further processed through a 37°C incubation with 0.2% Collagenase Type 1 (Worthington Biochemical Corporation) in DMEM-F12+ 10% FBS for 1 ½ hour. Erythrocytes were removed through cell lysing in ACK lysing buffer (Thermo Fisher Scientific). One 10 mm biopsy punch gave ~2.0 × $10^6$ cells. 1 million cells were blocked with TruStain Fc Block (anti-mouse CD16/32; BioLegend). Cells were stained for 30 min with antibodies specific to CD45-FITC (1:100; BioLegend), F4/80-PE (1:60; BD Biosciences), CD206-BV711 (1:100; BioLegend), CD64-PE-Cy7 (1:60; BioLegend), CD11b-BV395 (1:200; BD Biosciences), CD11c-BV421 (1:100; BD Biosciences), Ly6G-APC-Cy7 (1:60; BD Biosciences), IA/IE (MHCII)-PerCP/cy5.5 (1:125; BD Biosciences), CD301b-AF647 (1:100; BioLegend), Ly6C-AF700 (1:60; BD Biosciences). The stained cells were analyzed by flow cytometry.

## scRNA seq

Immune cells were obtained from 8-wk-old C57/Bl6 mice purchased from Jackson Laboratory (stock # 000664). The wound biopsy was performed as previously described. TGM/PBS (with 1.5% carboxymethylcellulose) was administered daily at 500 ng/50 $\mu$l under the Tegaderm. Mice from each treatment group (n = 3/group) were harvested on day 3. Excised skin was placed in 0.5% Dispase II solution (Roche) and placed at 4°C overnight. The following day, the digested skin was minced and further processed through a 37°C incubation with 0.2% Collagenase Type 1 (Worthington Biochemical Corporation) in DMEM-F12+ 10% FBS for 1 ½ hour. Erythrocytes were removed through cell lysing in ACK lysing buffer (Thermo Fisher Scientific). Cells were blocked with TruStain Fc Block (anti-mouse CD16/32; BioLegend). Cells from each sample were stained individually for 30 min with a master mix containing the antibodies specific to CD45-FITC (1:100; BioLegend) and a TotalSeq Hashtags Antibody marker (BioLegend). Each of the three biological replicates that were treated with PBS vehicle treated control or TGM was labeled with one of three unique TotalSeq Hashtags Antibody markers (BioLegend). The samples were individually sort-purified

by gating CD45$^+$ cells and then each treatment group was pooled with an even number of cells from each sample. The two-pooled samples (PBS and TGM) were then submitted to Rutgers genomics for further single-cell analysis using 10X genomics. For analysis, individual samples could be detected and further analyzed using the unique hashtag identifier. Briefly, as previously described (69), the counts were normalized and then compared between the two treatment groups to look at the differential gene expression using DESeq2.

### Histological analyses

A scalpel was used to cut the 10 mm punch wound down the midline to obtain the center of the wound. One wound half was mounted in optimum cutting temperature (O.C.T.) compound (Tissue-Tek, Sakura). 5–12 thick $\mu$m cryosections were cut on a cryostat using the first few several slices for analysis to ensure the wound center was used. The sections were air dried for 30 min to an hour and fixed in cold 99.5% histological grade Acetone (Sigma-Aldrich) for 15 min and then immunostained as previously described (69). Briefly, following fixation, tissue sections were blocked with 10% Rat Serum (Abcam) and 1% CD16/32 Fc Blocking antibody (BioLegend) in PBS for 1 h. They were then stained overnight at 4°C with the following antibodies: $\alpha$SMA for myofibroblast, AF488 1:100; BioLegend. Following primary staining, slides were stained with DAPI and then coverslips were applied using Prolong Gold Antifade Reagent (Life Technologies). Images were obtained through the Leica fluorescent microscope. Control images were used to normalize the fluorescence intensity of the photos taken for each sample.

The other half of the cut skin was used for H&E and picrosirius red analysis. The skin was formalin-fixed and then paraffin-embedded. 5 $\mu$m slides were used to stain for H&E and picrosirius red. The first several sections were again used to obtain the wound center for analysis. Wound and wound edge thickness were measured through ImageJ software. Collagen percentage was also analyzed using the MRI Fibrosis Tool plugin through ImageJ (44, 70).

### Protein analysis of serous discharge

For protein analysis of the serous discharge, 50 $\mu$l of sterile PBS was added to the exudate of each wound, before removing the Tegaderm, and then taken up. The additional liquid improved the ability to perform a full collection of the exudate. Since the volume from each mouse was small, each treatment group was pooled providing around 250 $\mu$l of solution from each treatment. The samples were spun to collect the cells. The supernatant was replaced by an additional 1 ml of sterile PBS. The protein concentration was obtained through a NanoDrop. The protein was concentrated using a 0.5 Centrifugal Filter Unit (Amicon) and then run on a Precast Protein Gel (456-1103; Bio-Rad) (71). Each well was loaded with 30 $\mu$l of a solution containing 25 $\mu$l of the protein and Pierce 5X Lane Marker Sample (Thermo Fisher Scientific). Three half dilutions of the protein, both PBS vehicle treated control and TGM treated, were made, denatured in 5% of 2-Mercaptoethanol, and then loaded into the gel which was run at 100 V in a 1X Tris–Glycine buffer. The gel was fixed in 50% methanol and 10% of acetic acid, stained overnight in Coomassie Blue Stain, and then de-stained the next day (72). The prepped gel was brought to the Center for Advanced Proteomics

Research (Rutgers Biomedical and Health Science New Jersey Medical School) for further protein analysis by LC-MS/MS.

### Bone marrow-derived macrophage isolation and culture

Bone marrow-derived monocytes were isolated from the femur and tibia of 8-wk-old C57/Bl6 mice (n = 3) as previously described (73). Erythrocytes were removed through cell lysing in ACK lysing buffer (Thermo Fisher Scientific) and resuspended in DMEM:F12 LCM lymphocyte growth media, which was obtained from L929 conditioned culture media and prepared and treated for in vitro differentiation as previously described (74). Briefly, cells were plated at 500,000 cells/well in a six-well plate. On day 6, the matured macrophages were stimulated for 16 h. All cells received MCSF (50 ng/ml; R&D Systems). Both recombinant LPS (100 ng/ml) and recombinant IFNy (10 ng/ml; Gibco) were provided to cells for differentiation to classical macrophages. Recombinant IL-4 (25 ng/ml; PeproTech) and IL-13 (25 ng/ml; R&D Systems) were provided to cells for alternatively activated macrophage differentiation. After incubation, the macrophages were trypsinized and scraped out of the well, centrifuged at 200$g$ for 10 min, and then resuspended in fresh lymphocyte media. The cells were processed for flow cytometry as described in the earlier methods. Cells were stained for 30 min with antibodies specific to CD45-FITC (1:100; BioLegend), F4/80-PE (1:60; BD Biosciences), CD206-BV711 (1:100; BioLegend), CD64-PE-Cy7 (1:60; BioLegend), CD11b-BV395 (1:100; BD Biosciences), CD11c-BV421 (1:100; BD Biosciences), Ly6C-AF700 (1:60; BD Biosciences). The stained cells were analyzed by flow cytometry.

### Statistical analyses

Data were analyzed with Prism 9 (GraphPad) and reported as mean ± SE. One-way ANOVA and protected $t$ tests were used to assess the difference between multiple groups at each time point. Comparisons between two groups were analyzed through a $t$ test. Each in vivo timepoint study or antibody study was replicated two or more times. A $P$-value of <0.05 was considered significant.

# Data and Materials Availability

All data are available in the main text or the supplementary materials. scRNAseq data were deposited into the Gene Expression Omnibus database under the accession number GSE272815.

# Supplementary Information

# Acknowledgements

National Institute of Health T32 Training Grant 5T32AI125185-04 (WC Gause) National Institute of Health R01 Award 1R01AI131634 (WC Gause) Wellcome Trust through an Investigator Award to RM Maizels 106122 and 219530 (RM

Maizels) Wellcome Trust core-funded Wellcome Centre for Integrative Parasitology 104111 (RM Maizels).

## Author Contributions

KE Lothstein: conceptualization, investigation, visualization, methodology, and writing—original draft, review, and editing.
F Chen: visualization, methodology, and writing—review and editing.
P Mishra: visualization, methodology, and writing—review and editing.
DJ Smyth: methodology and writing—review and editing.
W Wu: methodology and writing—review and editing.
A Lemenze: methodology and writing—review and editing.
Y Kumamoto: methodology and writing—review and editing.
RM Maizels: conceptualization, supervision, funding acquisition, investigation, visualization, methodology, and writing—review and editing.
WC Gause: conceptualization, supervision, funding acquisition, investigation, visualization, methodology, and writing—original draft, review, and editing.

## Conflict of Interest Statement

Rutgers, the State University of New Jersey has submitted a patent application on behalf of WC Gause, F Chen, Zhugong Liu, and P Mishra, which has been issue as Patent No. US 9,931,361 B2.

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
