## [Reviewer comments · Life Science Alliance]

Life Science Alliance

Helminth protein enhances wound healing by inhibiting fibrosis and promoting tissue regeneration

Katherine Lothstein, Fei Chen, Pankaj Mishra, Danielle Smyth, Wenhui Wu, Alexander Lemenze, Yosuke Kumamoto, Rick Maizels, and William Gause

DOI: <https://doi.org/10.26508/lsa.202302249>

Corresponding author(s): *William Gause, Rutgers University-Newark*

Review Timeline:

Submission Date:	2023-07-01
Editorial Decision:	2023-08-17
Revision Received:	2024-05-08
Editorial Decision:	2024-05-10
Revision Received:	2024-08-04
Accepted:	2024-08-05

Transaction Report:

August 17, 2023

Re: Life Science Alliance manuscript #LSA-2023-02249-T

Prof. William c Gause
Rutgers - New Jersey Medical School
185 South Orange Ave
Newark, NJ 7101

Dear Dr. Gause,

Thank you for submitting your manuscript entitled "Protein secreted by intestinal helminth enhances skin wound healing by inhibiting fibrosis and promoting tissue regeneration" to Life Science Alliance. The manuscript was assessed by expert reviewers, whose comments are appended to this letter. We invite you to submit a revised manuscript addressing the Reviewer comments.

Thank you for this interesting contribution to Life Science Alliance. We are looking forward to receiving your revised manuscript.

Sincerely,

B. MANUSCRIPT ORGANIZATION AND FORMATTING:

Reviewer #1 (Comments to the Authors (Required)):

This paper explores the use of a parasitic helminth derived TGF-beta mimic (TGM) in skin wound healing in the mouse. The paper describes a methodology whereby the daily treatment with TGM after a full skin biopsy wound was shown to accelerate wound closure compared to control treatment. Moreover, the treatment also promoted wound regeneration compared to controls. Overall the treatment convincingly showed that the parasite derived TGF-beta mimic offers strong potential as a treatment for skin wounds which promoted both tissue regeneration and wound closure. The data on the capacity of the compound to promote accelerated wound closure was clear as was the capacity to show tissue regeneration e.g. by collagen re-organisation and the formation of hair follicles. The data also suggested that the activity was operating through the TGF-beta receptor binding domains. There was also a series of experiments defining and characterising the re-programming of macrophages following TGM treatment both in vitro and in vivo. The data presented are derived from carefully carried out experiments and rigorous analysis. For the stage of the work the research is at I do not think any further experimental work is required. There is much more to do, of course, but the data stands on its own as presented. There are some points that I would like the authors to consider and perhaps incorporate into the discussion. No comment has been made as to the use of TGF-beta instead of TGM in the studies proposed. What would the authors anticipate to see, comparable response or distinct? I would expect that these kinds of experiments have been undertaken before by themselves or by others? Finally, as TGM is a parasite derived molecule, do the treated animals make antibodies against the TGM which may eventually reduce its efficacy? The data presented is clear cut but the model used is fairly rapid and the mouse does not present the opportunity to examine chronic wounds, where the opportunity to mount an anti-TGM response could potentially occur.

Reviewer #2 (Comments to the Authors (Required)):

The manuscript by Lothstein et al. examined the effect of TGF- mimic (TGM) isolated from parasitic worms on wound healing. The authors demonstrated that daily treatment with this compound accelerated wound closure, and promoted re-epithelialization, and hair follicle regeneration. Analysis of the immune response over the course of the wound healing process showed marked changes in the kinetics of macrophage accumulation in the tissue between groups with a faster accumulation of these cells in the TGM group. The maturation stage of wound healing is associated with alternatively associated macrophages. Surprisingly, the authors observed a reduced frequency of CD206+ cells, which they followed up with scRNAseq to better understand heterogeneity within the macrophage population. The data revealed that despite the downregulation of CD206, TGM does in fact induce a M2-like activation of macrophages at the wound site.

Overall, this is a clearly written and study that is of interest to the scientific community given its translational potential. The data are sound and consistent but there are a few issues listed below that should be addressed.

Major comments

1. In the abstract, the author state that "skin wound healing typically results in fibrosis and scarring" which seems to be an exaggeration. One would argue that most skin wounds in healthy individuals result in wound closure and tissue repair. Fibrosis is a possible pathogenic complication that can occur but is not the typical outcome.
2. The authors only provided frequencies of the various immune populations in figure 5. Absolute cell numbers would also be informative. At present, it is unclear whether the reduced frequency of tissue-resident macrophages at the early time points of wound healing is due to the loss/egress of these cells or the influx of monocyte-derived macrophages into the tissue.
3. The authors stated in the text describing figure 5A that "in the PBS vehicle-treated control group, the macrophage population (CD11b+ F4/80+ CD64+), as a percentage of all CD45+ cells, was increased at 1 to 2 days post-injury". This statement is not supported by the data shown. There is a large drop in macrophage frequency on day 2. The text should be corrected accordingly.
4. While the scRNAseq data strongly suggest that macrophages do acquire M2-like phenotype despite downregulation of

CD206+, validation by flow cytometry is necessary. Changes in gene expression don't always translate into protein. Furthermore, analysis of macrophage frequencies over the course of wound healing using alternative markers for M2 macrophages (Arg1 etc) would also allow for a better understanding of the immune response after TGM treatment. This data would substitute the analysis of the frequency of CD206+ macrophages in figure 5, which is not very informative as this marker gets downregulated in the treatment group. In general, the downregulation of CD206 after TGM treatment is a good example of why relying on a single surface marker might be misleading, especially in vivo.

5. Did applying the TGM treated at the wound side have any systemic effect? Have the authors looked in the draining lymph nodes? Were there any changes in Treg frequencies or absolute numbers?

Minor comments

1. The authors should use consistent nomenclature when referring to their control group in text, figures, and figure legends. It varies between PBS, PBS vehicle, and Ctl (e.g. sup fig 1d,e). It makes it difficult to discern whether the same vehicle control was used in all experiments.

2. Can the authors comment on or speculate why the administration of a single dose of TGM on day 0 (sup. Fig. 1d) resulted in slower wound healing than control?

3. There is a lack of consistency in the type of bar plots in the figures: some bar plots have just the average data, while other ones show individual samples. Generally, showing individual replicates would be preferable.

4. In figure 5A, it would be helpful for the reader if the authors specified in the legend (underneath the symbol for the different cell types) that the bars represent the frequency of open wounds). While it is implied, it is not immediately clear.

5. The small text at the bottom corners of the flow plots in Figure 5F is unreadable.

6. A discussion of how the TGM might directly or indirectly promote macrophage accumulation would be helpful. It would be good to put the data in the context of the fact that TGFB has been previously shown to promote M2 polarization.

7. Can the authors clarify why only male mice were used in this study? Was there no difference in females?

Reviewer #3 (Comments to the Authors (Required)):

The present study submitted by Lothstein et al describes the wound healing effects of a TGF-beta mimic protein from the excretory secretory products of the GI nematode parasite *H. polygyrus* the authors show that repeated administration of a 500 ng bolus of TGM accelerates in vivo tissue repair in a punch biopsy model. Enhanced serous fluid containing Haptoglobin and Thrombospondin were detected, the latter of which is a known activator of mammalian TGF-beta. Further, TGM accelerated re-epithelialization and hair follicle regeneration with the organization of collagen fibrils consistent with a non-fibrotic wound healing cascade. Overall, this is a solid study and may lead to novel avenues for understanding and potentially treating the wound healing process. The data shown are robust and appropriate statistical tests were performed.

Comments: In Fig S1, the dose responses show indication of several doses where TGM actually delayed wound closure. Can the authors comment on why this occurred.

Have the authors confirmed that these effects of TGM were found in mice of distinct genetic backgrounds? Are any of these effects dependent upon signaling through TLRs? Indeed, while significant, some of differences observed with enhanced wound edge thickness and leading edge thickness were moderate and seemed to only occur at a distinct time-point.

Was the enhanced hair follicle regeneration dependent upon endogenous Foxp3+ Tregs? What impact did TGM have on the skin stem cell compartment?

Is the pro-regenerative effect of TGM dependent on tissue macrophages? Can the authors inhibit the effects of TGM by application of clodronate liposomes?

The purity and LPS content of TGM should be reported.

Reviewer #1 (Comments to the Authors (Required)):

- This paper explores the use of a parasitic helminth derived TGF-beta mimic (TGM) in skin wound healing in the mouse. The paper describes a methodology whereby the daily treatment with TGM after a full skin biopsy wound was shown to accelerate wound closure compared to control treatment. Moreover, the treatment also promoted wound regeneration compared to controls.
- Overall the treatment convincingly showed that the parasite derived TGF-beta mimic offers strong potential as a treatment for skin wounds which promoted both tissue regeneration and wound closure.
- The data on the capacity of the compound to promote **accelerated wound closure was clear as was the capacity to show tissue regeneration e.g. by collagen re-organisation and the formation of hair follicles**. The data also suggested that the activity was operating through the TGF-beta receptor binding domains.
- There was also a series of experiments defining and characterising the re-programming of macrophages following TGM treatment both in vitro and in vivo.
- The data presented are derived from carefully carried out experiments and rigorous analysis. **For the stage of the work the research is at I do not think any further experimental work is required**. There is much more to do, of course, but the data stands on its own as presented.
- There are some points that I would like the authors to consider and perhaps incorporate into the discussion.

1. No comment has been made as to the use of TGF- beta instead of TGM in the studies proposed. What would the authors anticipate to see, comparable response or distinct?

1. I would expect that these kinds of experiments have been undertaken before by themselves or by others?

- Preliminary work comparing the bioactivity of TGM to TGF-beta has shown that a comparable amount of TGF-B is beyond the limit of physiological significance. TGM has much greater bioactivity level at a smaller dosing range than recombinant TGF-beta and therefore, we were unable to use the TGF-beta comparison in our in vivo studies.

2. Finally, as TGM is a parasite derived molecule, do the treated animals make antibodies against the TGM which may eventually reduce its efficacy? The data presented is clear cut but the model used is fairly rapid and the mouse does not present the opportunity to examine chronic wounds, where the opportunity to mount and anti-TGM response could potentially occur.

- Tests have previously been performed to show that TGM is non immunogenic. Refer to the following paper in which TGM is used in a model of allograft rejection: C. J. C. Johnston *et al.*, A structurally distinct TGF-beta mimic from an intestinal helminth parasite potently induces regulatory T cells. *Nat Commun* **8**, 1741 (2017).

Reviewer #2 (Comments to the Authors (Required)):

- The manuscript by Lothstein et al. examined the effect of TGF- β mimick (TGM) isolated from parasitic worms on wound healing. The authors demonstrated that daily treatment with this compound accelerated wound closure, and promoted re-epithelialization, and hair follicle regeneration. Analysis of the immune response over the course of the wound healing process showed marked changes in the kinetics of macrophage accumulation in the tissue between groups with a faster accumulation of these cells in the TGM group. The maturation stage of wound healing is associated with alternatively associated macrophages. Surprisingly, the authors observed a reduced frequency of CD206+ cells, which they followed up with scRNAseq to better understand heterogeneity within the macrophage population. The data revealed that despite the downregulation of CD206, TGM does in fact induces a M2-like activation of macrophages at the wound site.
- Overall, this is a clearly written and study that is of interest to the scientific community given its translational potential. The data are sound and consistent but there are a few issues listed below that should be addressed.
- Major comments
 1. **In the abstract, the author state that "skin wound healing typically results in fibrosis and scarring" which seems to be an exaggeration. One would argue that most skin wounds in healthy individuals result in wound closure and tissue repair. Fibrosis is a possible pathogenic complication that can occur but is not the typical outcome.**
 - Line 23 – Edited to “Skin wound healing due to full thickness wounds typically results in fibrosis and scarring
 2. **The authors only provided frequencies of the various immune populations in figure 5. Absolute cell numbers would also be informative. At present, it is unclear whether the reduced frequency of tissue-resident macrophages at the early time points of wound healing is due to the loss/egress of these cells or the influx of monocyte-derived macrophages into the tissue.**
 - Yes, we would have liked to provide total cell count, however, due to the technical manner through which the biopsies were obtained, it was difficult to maintain complete consistency between the sizes of the biopsies when collected. As a result, the total cell number would not be accurate and comparable between samples. Therefore, it was decided that frequency would be the best way to normalize the results.
 3. **The authors stated in the text describing figure 5A that "in the PBS vehicle-treated control group, the macrophage population (CD11b+ F4/80+ CD64+), as a percentage of all CD45+ cells, was increased at 1 to 2 days post-injury". This statement is not supported by the data shown. There is a large drop in macrophage frequency on day 2. The text should be corrected accordingly.**
 - Line 214: Edited to “....percentage of all CD45+ cells, was decreased at 1 to 2 days post-injury, however; the percentage began to increase and peaked between 5 to 7 days...”
 4. **While the scRNAseq data strongly suggest that macrophages do acquire M2-like phenotype despite downregulation of CD206+, validation by flow cytometry is**

necessary. Changes in gene expression don't always translate into protein. Furthermore, analysis of macrophage frequencies over the course of wound healing using alternative markers for M2 macrophages (Arg1 etc) would also allow for a better understanding of the immune response after TGM treatment. This data would substitute the analysis of the frequency of CD206+ macrophages in figure 5, which is not very informative as this marker gets downregulated in the treatment group. In general, the downregulation of CD206 after TGM treatment is a good example of why relying on a single surface marker might be misleading, especially in vivo.

- We appreciate your insight and agree. For the purpose of this study, our primary focus was first to understand the wound healing capabilities of TGM. We further provided a preliminary investigation of the potential mechanism of action, which included the gene expression analyses. We now include a statement in lines 278-280 that:
 - “It will be important in future studies to examine whether these changes observed in expression of M2-associated genes also occur at the protein level.”
5. **Did applying the TGM treated at the wound side have any systemic effect? Have the authors looked in the draining lymph nodes? Were there any changes in Treg frequencies or absolute numbers?**
- While broad systemic effects were not analyzed and we did not analyze the lymph nodes, we did look at the peritoneum. The data provided similar results as to what was observed in the skin including increased M2 populations. However, the results observed were much greater in the skin. We plan to perform further studies on the periphery in the future.

- **Minor comments**

1. **The authors should use consistent nomenclature when referring to their control group in text, figures, and figure legends. It varies between PBS, PBS vehicle, and Ctl (e.g. sup fig 1d,e). It makes it difficult to discern whether the same vehicle control was used in all experiments.**
 - This has been corrected and we thank the reviewers for noting this.
2. **Can the authors comment on or speculate why the administration of a single dose of TGM on day 0 (sup. Fig. 1d) resulted in slower wound healing than control?**
 - Based on the results, we did not see any significant difference between one dose and the control. There was no significant difference in the wound closure rate of the various doses vs the PBS treated control.
3. **There is a lack of consistency in the type of bar plots in the figures: some bar plots have just the average data, while other ones show individual samples. Generally, showing individual replicates would be preferable.**
 - I have changed the graphs to match and include individual samples.
4. **In figure 5A, it would be helpful for the reader if the authors specified in the legend (underneath the symbol for the different cell types) that the bars represent the frequency of open wounds). While it is implied, it is not immediately clear.**
 - We thank the reviewer for noting this and have added to the legend in Fig. 5a.
5. **The small text at the bottom corners of the flow plots in Figure 5F is unreadable.**
 - The small text has been increased so that it is now legible.

6. **A discussion of how the TGM might directly or indirectly promote macrophage accumulation would be helpful. It would be good to put the data in the context of the fact that TGFB has been previously shown to promote M2 polarization.**
 - A brief discussion has been added.
 - Line 308: TGF- β is known to play an important role in wound healing and has previously been shown to promote M2 macrophage activation (ref). Our findings now indicate that TGM also promotes activation and proliferation of tissue-repair associated M2 macrophages through either direct or indirect mechanisms.
7. **Can the authors clarify why only male mice were used in this study? Was there no difference in females?**
 - There has been some documented morphologic differences between male and female mice between the thickness of the dermis which could impact the wound healing response. To prevent this variable, we decided to specifically work with male mice, only, to maintain consistency. We now reference this in the methods section on line 382.

Reviewer #3 (Comments to the Authors (Required)):

- The present study submitted by Lothstein et al describes the wound healing effects of a TGF-beta mimic protein from the excretory secretory products of the GI nematode parasite *H. polygyrus* the authors show that repeated administration of a 500 ng bolus of TGM accelerates in vivo tissue repair in a punch biopsy model. Enhanced serous fluid containing Haptoglobin and Thrombospondin were detected, the latter of which is a known activator of mammalian TGF-beta. Further, TGM accelerated re-epithelialization and hair follicle regeneration with the organization of collagen fibrils consistent with a non-fibrotic wound healing cascade.
- Overall, this is a solid study and may lead to novel avenues for understanding and potentially treating the wound healing process. The data shown are robust and appropriate statistical tests were performed.
- Comments:
 1. **In Fig S1, the dose responses show indication of several doses where TGM actually delayed wound closure. Can the authors comment on why this occurred.**
 - In FigS1, it may seem that there are TGM doses that are leading to slower wound closure than the control. However, based on statistical analysis, there

was no significant difference between these lower doses and the PBS treated control. This demonstrates that the lower doses are giving no effect, rather than a slow response, compared to the control. We have now emphasized this in a comment in the Results on line 104.

- 2. Have the authors confirmed that these effects of TGM were found in mice of distinct genetic backgrounds?**
 - The work was only performed in WT Bl.6 male mice with no distinct disease state. Future work could further delve into various genetic backgrounds but the focus of this study was to first identify the possible therapeutic potential of this molecule in an experimental model.
- 3. Are any of these effects dependent upon signaling through TLRs?**
 - We did not test this possibility in this study, but it is certainly an area that we should consider in future experiments. For that during this study but will consider this analysis in future work
- 4. Indeed, while significant, some of differences observed with enhanced wound edge thickness and leading edge thickness were moderate and seemed to only occur at a distinct time-point.**
 - Yes, the thickening of the wound edges does correlate with a specific timepoint in the normal process of wound healing. What our analysis showed is that this is exaggerated in our treated group which may account for the accelerated wound closure that we observed.
- 5. Was the enhanced hair follicle regeneration dependent upon endogenous Foxp3+ Tregs?**
 - Previously published data has shown that Tregs do play a role in hair follicle regrowth by enhancing the proliferation of the hair follicle stem cells. With our study, we cannot yet state whether the regeneration is due to the effects of the TGM on Tregs or through another mechanism but we will consider this for future studies and now include a brief discussion of this possibility in the text on line 304.
- 6. What impact did TGM have on the skin stem cell compartment?**
 - While we recognize the importance of the stem cells located in the hair follicles, we did not yet look at this population and its response to TGM. We will certainly consider this in future studies.
- 7. Is the pro-regenerative effect of TGM dependent on tissue macrophages? Can the authors inhibit the effects of TGM by application of clodronate liposomes?**
 - The wound healing system usually involves both tissue derived as well as recruited macrophages at different stages of the wound healing process. We have noted that TGM appears to impact both types of macrophages based on some of the increased presence of migratory markers on the macrophage population in the wound bed. Research has indicated that these two populations play different roles in the process. This will be a focus of future studies
- 8. The purity and LPS content of TGM should be reported.**
 - We now specifically report this on line 374.

May 10, 2024

RE: Life Science Alliance Manuscript #LSA-2023-02249-TR

Prof. William c Gause
Rutgers University-Newark
185 South Orange Ave
Newark, NJ 7101

Dear Dr. Gause,

Thank you for submitting your revised manuscript entitled "Helminth protein enhances wound healing by inhibiting fibrosis and promoting skin regeneration". We would be happy to publish your paper in Life Science Alliance pending final revisions necessary to meet our formatting guidelines.

- please be sure that the authorship listing and order is correct
- please upload your main manuscript text as an editable doc file
- please upload all figure files individually, including the supplementary figure files; all figure legends should appear only in the main manuscript file, not with the figures
- please add your main and supplementary figure legends to the main manuscript text after the references section. --please name the sections 'Figure legends' and 'Supplementary figure legends.' There should be only one set of legends for each set of figures.
- please add the Twitter handle of your host institute/organization as well as your own or/and one of the authors in our system
- please add ORCID ID for the corresponding author -- you should have received instructions on how to do so
- title in the manuscript and system must match
- please use the [10 author names et al.] format in your references (i.e., limit the author names to the first 10)
- please add a Conflict of Interest statement to your main manuscript text
- please consult our manuscript preparation guidelines <https://www.life-science-alliance.org/manuscript-prep> and make sure your manuscript sections are in the correct order
- please either update the contributions in our system and the Author Contributions section of the manuscript for Danielle Smyth, as the currently listed contribution do not constitute authorship, or let us know if the author should be removed.
- please provide accession information for the RNA-seq data in the Data Availability statement

FIGURE CHECKS

- please add scale bars to Figure 1A and S3B

A. FINAL FILES:

B. MANUSCRIPT ORGANIZATION AND FORMATTING:

Sincerely,

August 5, 2024

RE: Life Science Alliance Manuscript #LSA-2023-02249-TRR

Prof. William c Gause
Rutgers University-Newark
185 South Orange Ave
Newark, NJ 7101

Dear Dr. Gause,

Thank you for submitting your Research Article entitled "Helminth protein enhances wound healing by inhibiting fibrosis and promoting tissue regeneration". It is a pleasure to let you know that your manuscript is now accepted for publication in Life Science Alliance. Congratulations on this interesting work.

DISTRIBUTION OF MATERIALS:

Again, congratulations on a very nice paper. I hope you found the review process to be constructive and are pleased with how the manuscript was handled editorially. We look forward to future exciting submissions from your lab.

Sincerely,
